# Whole-body protein kinetics in critically ill patients during 50 or 100% energy provision by enteral nutrition: A randomized cross-over study

**Martin Sundström Rehal** [1,2]*, **Felix Liebau** [1,2], **Jan Wernerman** [2], **Olav Rooyackers** [1,2]

**1** Department of Perioperative Medicine and Intensive Care (PMI), Karolinska University Hospital Huddinge, Stockholm, Sweden, **2** Division of Anesthesia and Intensive Care, Department of Clinical Science, Intervention and Technology (CLINTEC), Karolinska Institutet, Stockholm, Sweden

\* martin.sundstrom@gmail.com

**Data Availability Statement:** A comprehensive report of individual patient characteristics has been omitted from the database to avoid potential

## Abstract

### Background

Enteral nutrition (EN) is a ubiquitous intervention in ICU patients but there is uncertainty regarding the optimal dose, timing and importance for patient-centered outcomes during critical illness. Our research group has previously found an improved protein balance during normocaloric versus hypocaloric parenteral nutrition in neurosurgical ICU patients. We now wanted to investigate if this could be demonstrated in a general ICU population with established enteral feeding, including patients on renal replacement therapy.

### Methods

Patients with EN >80% of energy target as determined by indirect calorimetry were randomized to or 50% or 100% of current EN rate. After 24 hours, whole-body protein kinetics were determined by enteral and parenteral stable isotope tracer infusions. Treatment allocation was then switched, and tracer investigations repeated 24 hours later in a crossover design with patients serving as their own controls.

### Results

Six patients completed the full protocol. During feeding with 100% EN all patients received >1.2 g/kg/day of protein. Mean whole-body protein balance increased from -6.07 to 2.93 μmol phenylalanine/kg/h during 100% EN as compared to 50% (p = 0.044). The oxidation rate of phenylalanine was unaltered (p = 0.78).

### Conclusions

It is possible to assess whole-body protein turnover using a stable isotope technique in critically ill patients during enteral feeding and renal replacement therapy. Our results also suggest a better whole-body protein balance during full dose as compared to half dose EN. As

identification of research subjects. All other data constituting the minimal dataset for replicating the results presented in the manuscript are available from the Swedish National Data Service database (https://doi.org/10.5878/b1e8-fg58).

**Funding:** This study was funded by grants to JW from Stockholm City Council (https://www.sll.se/), grant #563170, and the Swedish Research Council (https://www.vr.se/english.html), grant #04210. The funders had no role in study design, data collection and analysis, decision to publish, or preparation of the manuscript.

**Competing interests:** We have read the journal's policy and the authors of this manuscript have the following competing interests: JW and OR have given paid lectures about nutrition in the ICU for Nestlé, Nutricia and Fresenius Kabi. OR is a consultant for Fresenius-Kabi. FL has received a speaking fee from Baxter. MSR has no competing interests to declare. This does not alter our adherence to PLOS ONE policies on sharing data and materials.

**Abbreviations:** ARDS, Acute respiratory distress syndrome; BW, Body weight; CVVHD, Continuous veno-venous hemodialysis; CRRT, Continuous renal replacement therapy; EE, Energy expenditure; EN, Enteral nutrition; $FiO_2$, Fraction of inspired oxygen; ICU, Intensive care unit; MPE, Molar percentage excess; Neuro ICU, Neurosurgical ICU; Phe, Phenylalanine; Ra, Rate of appearance; Rd, Rate of disappearance; RCT, Randomized controlled trial; SAPS, Simplified acute physiology score; SOFA, Sequential organ failure assessment; TPN, Total parenteral nutrition.

the sample size was smaller than anticipated, this finding should be confirmed in larger studies.

## Introduction

The role of energy and protein delivery during critical illness remains unclear. In recent years several large randomized controlled trials (RCTs) have failed to demonstrate a reduction in mortality from increased energy delivery during critical illness [1–3]. However, the effects of nutritional interventions on lean body mass preservation and functional recovery have not been investigated to the same extent. Results from several small RCTs have been conflicting [4–8] and currently there is insufficient data to conclude if energy and protein delivery within recommendations from clinical guidelines can ameliorate muscle loss in critically ill patients [9–11].

It is plausible to think that a nutritional intervention should have a positive effect on protein balance to be able to protect muscle mass. Our research group previously demonstrated that an increase in energy and protein delivery by total parenteral nutrition (TPN) from 50 to 100% of measured energy expenditure improves whole-body protein balance in neurosurgical ICU patients [12]. Two factors limit the generalization of this finding: the patients studied suffer from a specific pathophysiology different from the majority of ICU patients, and the use of TPN is relatively rare in current clinical practice [13]. Although enteral nutrition (EN) is the predominant route of feeding during critical illness, variable uptake from the gastrointestinal tract, splanchnic sequestration of nutrients and the hormonal response to enteral feeding impedes extrapolation of results from studies performed with exclusively parenteral nutrition. We therefore wanted to investigate the effects of 50% versus 100% delivery of energy targets by EN on whole-body protein balance (primary outcome measure) and plasma amino acid profile (secondary outcome measure) in a general ICU population.

## Methods

### Trial registration

This study was prospectively registered on 2014-05-08 at the Australian New Zealand Clinical Trials Registry, registration number ACTRN12614000476639, https://www.anzctr.org.au/Trial/Registration/TrialReview.aspx?id=366000&isReview=true.

### Ethics statement

This study was approved by the regional ethical committee in Stockholm county (DNR: 2016/76-31/4). Informed consent was obtained from all patients or the closest relative if the patient could not communicate his or her intentions. Information was provided both orally and in writing, clearly stating that the patient could withdraw from the study at any time.

### Patients

This study was conducted in a mixed medical-surgical ICU of a tertiary referral hospital (Karolinska University Hospital Huddinge, Stockholm, Sweden). The unit does not manage cardiothoracic/neurosurgical patients or patients on extracorporeal membrane oxygenation. Patients admitted to the ICU during the study period were screened for inclusion. Inclusion criteria were:

1. Invasive mechanical ventilation with an $FiO_2$ of ≤0.6.

2. ≥18 years of age.

3. ≥80% of target calories by enteral route as determined by indirect calorimetry.

   Patients were excluded if:

1. Informed consent could not be provided by the patient or next-of-kin.

2. Extubation, withdrawal of life support, termination of enteral tube feeding or transfer to another hospital/ward was anticipated during the study period.

3. No central venous or arterial catheters were available for blood sampling.

4. Volume resuscitation or blood transfusions were administered during the measurement periods.

Inclusion/exclusion (1,3/2,3) criteria listed above have been modified compared to the original study protocol in order to facilitate screening and recruitment of patients. Continuous renal replacement therapy (CRRT) was not an exclusion criterion if blood flow, dialysate flow and ultrafiltration rate remained constant during the sampling period and 2 hours prior. The routine CRRT modality in the unit is continuous veno-venous hemodialysis (CVVHD) with citrate anticoagulation. As the filtration rate of citrate was not measured within the context of this study, energy delivery from systemic uptake and metabolism was estimated from previous publications [14]. According to standard operating procedures at our unit, patients treated with CRRT during the study period also received an intravenous glutamine and alanine infusion (Dipeptiven, Fresenius Kabi) at a rate corresponding to 20 g/day during 50% EN and 40 g/day during 100% EN to compensate for loss of glutamine across the hemofilter. All aspects of care beyond the study protocol were determined by the attending physician and care team.

Patient recruitment and follow-up was conducted between 2016-12-14 and 2018-03-05 (Fig 1). 12 patients were enrolled in the study, but the rate of protocol violations was higher than expected (N = 6). In five cases the protocol was interrupted by clinical circumstances and in one case the nutrition protocol was violated due to unexpected feeding intolerance.

## Protocol

The study design was an experimental, cross-over randomized trial with patients serving as their own controls. After inclusion, patients were randomized to continuous enteral feeding at either 50% or 100% rate of ongoing EN. Treatment allocation was assigned by drawing numbers (1 = 100%, 2 = 50%) from sealed opaque envelopes arranged in random order by a computer-generated sequence (https://www.randomizer.org). Block randomization was not performed as this was not communicated to research staff. The individuals responsible for allocating treatment order did not take part in preparing the randomization process. Prior to starting the protocol, indirect calorimetry (Quark RMR, Cosmed, Rome, Italy) was performed to determine energy expenditure (EE) at baseline and ascertain that the current energy target was ≥80% of EE. At time $(T)_1 = 0$ EN rate was set to the initial level determined by treatment allocation. At $T_2 = 19$ hours, an enteral infusion of $1\text{-}^{13}C$-phenylalanine was added in parallel to the ongoing EN at a rate determined to provide a 30% enrichment (i.e. fraction of isotopically labeled compound) of total enteral phenylalanine. At $T_3 = 21$ hours, primed intravenous infusions of ring-$D_5$-phenylalanine (prime 2.94 μmol/kg, infusion 2.94 μmol/kg/h), 3,3-$D_2$-tyrosine (prime 1.64 μmol/kg, infusion 1.64 μmol/kg/h) and an intravenous bolus of ring-$D_4$-tyrosine (0.81 μmol/kg) were administered. At $T_4 = 23.45$ hours, four blood samples were drawn at five-minute intervals. A 15 minute sampling period was chosen over a 30 minute period specified in the

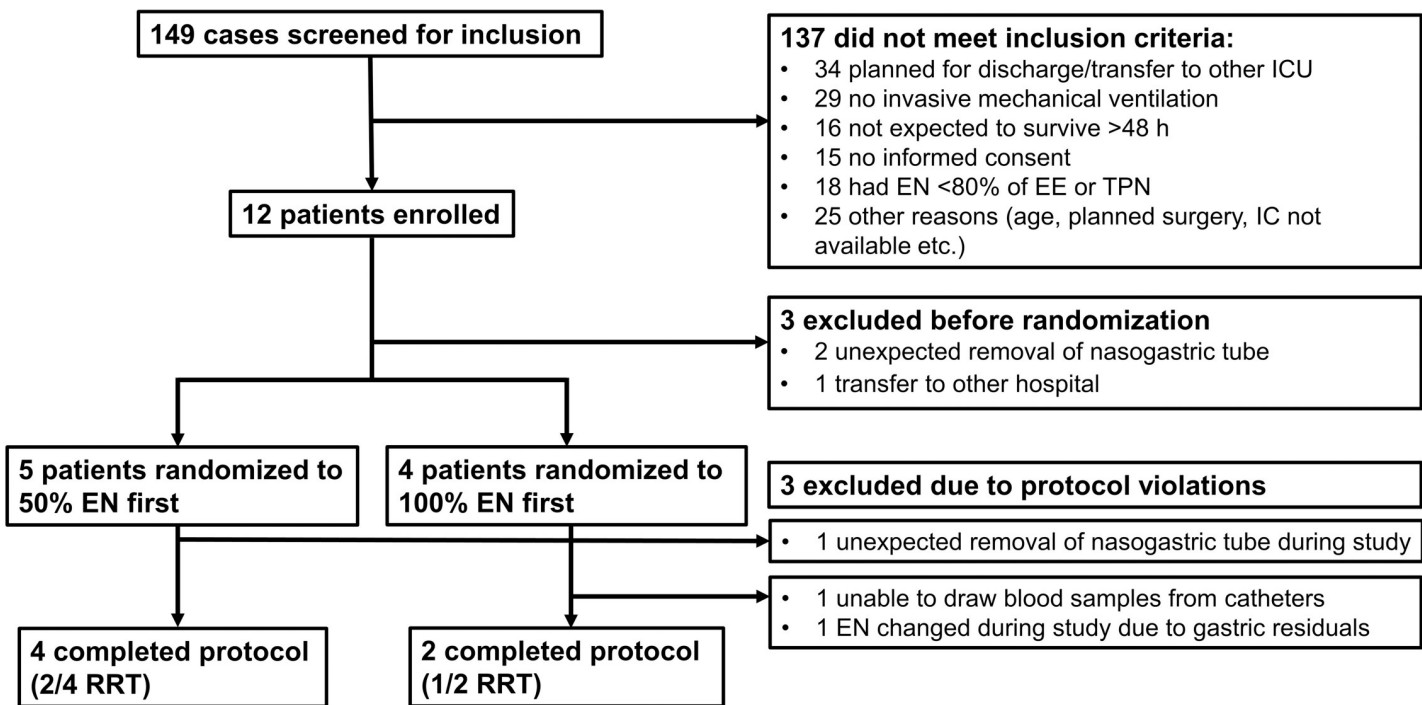

**Fig 1. CONSORT recruitment flowchart.** EN: Enteral nutrition; IC: Indirect calorimetry; ICU: Intensive care unit; RRT: Renal replacement therapy; TPN: Total parenteral nutrition.

original protocol as this had successfully been used in similar studies and reduced the interruption in patient care [15]. If the patient was receiving CRRT at the time, the effluent bag was emptied prior to the sampling period, the volume of effluent collected during sampling was measured and a sample of effluent was drawn to quantify filtration rate of amino acids across the hemofilter. Both the enteral and intravenous tracer infusions were then stopped, and a new indirect calorimetry was performed if possible. At $T_5 = 24$ hours, the rate of EN was changed to either 50% or 100% of baseline, depending on initial treatment allocation. At $T_6 = 43$ hours, the stable isotope infusions and sampling procedure was repeated at identical intervals to day one. If possible, a final determination of EE by indirect calorimetry was performed at the end of the protocol, and clinical data from the study period extracted from the patient data management system and electronic records. The protocol is illustrated in Fig 2.

Blood samples were collected in EDTA tubes, centrifuged to obtain plasma and frozen to -80 °C. Molar percent excess (MPE) of the tracers was determined from average values in the four blood samples drawn at the end the tracer protocol and measured by gas chromatography-mass spectrometry (Agilent N5973; Agilent, Kista, Sweden) as described in detail previously [16]. The last sample from each measurement period was also analyzed for serum urea (Urea kit on Indiko analyser, Thermo Fisher Scientific) and plasma free amino acid concentrations with high performance liquid chromatography (Alliance; Waters Corporation, Milford, MA, USA). Amino acid concentration is given as the sum of all measured amino acids. In addition, phenylalanine concentration was measured in the CRRT effluent.

## Calculations

Phenylalanine (Phe) rate of appearance ($Ra_{Phe}$) into the central compartment is calculated as:

$$Ra_{Phe} = i * (E_i / E_A) - i$$

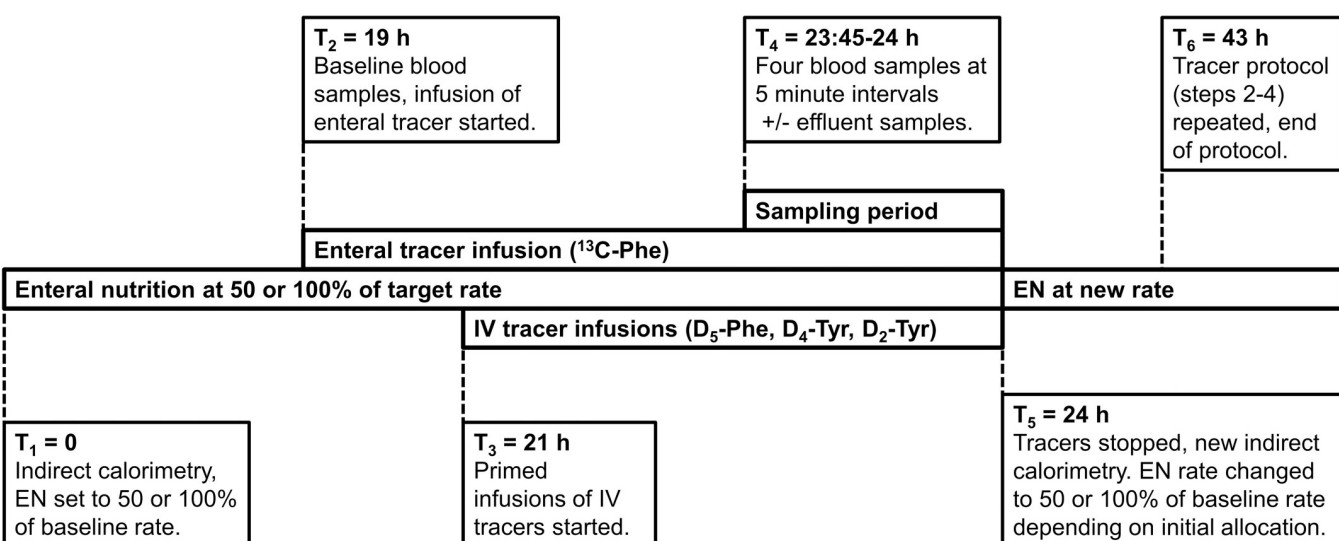

**Fig 2. Schematic illustration of study protocol (not to scale).** EN: Enteral nutrition; IV: intravenous; Phe: Phenylalanine; T: Time; Tyr: Tyrosine.

Where i: infusion rate of tracer (μmol/kg/h); $E_i$: Enrichment of infusate (molar percentage excess, MPE); $E_A$: Arterial enrichment (MPE). During a physiological steady state situation, Ra equals rate of disappearance (Rd).

Whole-body protein breakdown is estimated from $Ra_{Phe}$ minus exogenous (parenteral and enteral) Phe delivery. Enteral Phe is adjusted for splanchnic extraction by comparing appearance rates of the enteral ($^{13}$C-Phe) and parenteral ($D_5$-Phe) tracers in plasma:

$$\text{Splanchnic extraction fraction} = 1 - ((E_{A13C-Phe}/i_{13C-Phe})/(E_{A\ D5-Phe}/i_{D5-Phe}))$$

Whole-body protein synthesis is estimated from $Rd_{Phe}$ minus oxidation of Phe and the loss of Phe over the hemofilter. $\text{Oxidation}_{Phe}$ is estimated from the whole-body conversion of phenylalanine to tyrosine:

$$\text{Oxidation}_{Phe} = Ra_{Tyr}*((E_{A\ D4-Tyr})/(E_{A\ D5-Phe}))*(Ra_{Phe}/(i_{D5-Phe} + Ra_{Phe}))$$

Whole-body protein balance is calculated by the difference between protein synthesis and protein breakdown. The kinetic calculations and analytical methods used have been extensively described in previous publications [16–18].

## Statistics

Sample size calculations were based on data from a previous investigation by our research group with a similar study design [12]. The original power calculation for the study protocol was based on the number of subjects required to observe a change in the primary outcome corresponding to 75% of an anticipated standard deviation of 12.7 μmol Phe/kg/h, assuming a mean balance of -11.5 μmol Phe/kg/h. As a change to net neutral balance was considered more meaningful this was subsequently modified to detect a mean change in whole-body protein balance from -11.5 μmol Phe/kg/h to zero with α = 0.05 and β = 0.2, requiring 10 patients with complete data for the primary outcome. To provide margin for protocol violations due to unexpected circumstances the recruitment target was set to twelve patients. Significance testing was performed using a two-tailed Student's t-test for paired samples. Amino acid kinetics were assumed to be normally distributed from prior observations [12, 15, 19]. Descriptive data

**Table 1. General patient characteristics.**

|  | Admission diagnosis | Age | SAPS III | SOFA day 1 | SOFA day 2 | SOFA day 3 | ICU LoS study day 1 |
|---|---|---|---|---|---|---|---|
| Patient 1 | Septic shock | 54 | 108 | 10 | 10 | 10 | 5 |
| Patient 2 | Respiratory failure | 73 | 106 | 8 | 8 | 10 | 21 |
| Patient 3 | Septic shock | 56 | 70 | 3 | 1 | 1 | 28 |
| Patient 4 | Endocarditis | 28 | 45 | 8 | 8 | 7 | 30 |
| Patient 5 | Septic shock | 72 | 95 | 5 | 4 | 2 | 9 |
| Patient 6 | ARDS | 59 | 67 | 4 | 4 | 4 | 7 |

ARDS: Acute respiratory distress syndrome; ICU: Intensive care unit; LoS: Length of stay; SOFA: Sequential organ failure assessment; SAPS: Simplified acute physiology score.

and results corrected for body weight (BW) are adjusted using the same formula as applied in local clinical practice ($BW_{calc}$ = [Height (cm) - 100] + [Admission BW (kg)–[Height (cm)–100]]*0.33). All calculations and graphical presentations of data were performed in Prism 8.3 (GraphPad Software Inc, San Diego, USA).

## Results

The six patients (5 men, 1 woman) who completed the full protocol are characterized in Table 1. Information on anthropometric variables and nutritional therapy are provided in Tables 2 and 3. One patient received a 5% dextrose infusion on both study days, but no other parenteral nutrition was provided. Three patients received CVVHD with citrate anticoagulation and supplementary amino acids during the study period. Patients were primarily sedated with clonidine and morphine. 4/6 patients had 50% EN as their initial treatment allocation.

During 100% feeding all patients received ≥1.2 g/kg/day of protein. Mean whole-body protein balance increased from -6.1 µmol Phe/kg/h during 50% EN to +2.9 at 100% (p = 0.044). No statistically significant changes in any other kinetic parameters were observed (Table 4). The mean and individual values of kinetics during 50 and 100% EN are illustrated in Fig 3. Plasma free amino acids increased during 100% EN (p = 0.011) but there was no change in serum urea (p = 0.28). Plasma free amino acids did not differ between patients with/without CRRT and Dipeptiven supplementation (Fig 4). Complete plasma aminograms during 50 and 100% EN for individual patients are presented in Table 5. In the three patients with CRRT, loss of phenylalanine across the hemofilter was between 1.9–4.8 µmol/kg/h.

**Table 2. Anthropometric, nutritional and metabolic patient characteristics.**

|  | Baseline | 50% EN | 100% EN |
|---|---|---|---|
| BMI (kg/m$^2$) | 27.1 (19.5–40.7) |  |  |
| Energy intake (kcal/kg/day) |  | 16.3 (12.2–22.3) | 32.2 (24.5–44.5) |
| Protein intake (g/kg/day) |  | 0.89 (0.6–1.38) | 1.77 (1.20–2.77) |
| REE (kcal/kg/day) | 33.0 (26.3–44.6) | 32.5 (22.4–46.8) | 36.1 (33.2–38.5)* |
| Energy intake/REE (%) |  | 54 (42–64) | 91 (73–106)** |
| Respiratory quotient | 0.83 (0.70–0.88) | 0.76 (0.70–0.81) | 0.78 (0.72–0.83) |
| Energy deficit (Intake–REE, kcal/kg/day) |  | -16.2 (-24.5 - -10.2) | -5.1 (-9.2–2.1)** |
| Insulin dose (IU/h) |  | 1.58 (0–3.4) | 1.67 (0–3.0) |

All data presented as mean (range). BMI: Body mass index; REE: Resting energy expenditure; EN: Enteral nutrition.

*Two missing values where indirect calorimetry could not be performed due to extubation on day 2.

**Missing REE values (n = 2) imputed from baseline measurement.

**Table 3. Individual patient nutritional and metabolic characteristics.**

|  | EE* baseline | EE* 50% EN | EE* 100% EN | Enteral formula | CRRT | Dipeptiven | Kcal 50% | Kcal 100% | Protein** 50% | Protein** 100% |
|---|---|---|---|---|---|---|---|---|---|---|
| Patient 1 | 27.8 | 22.4 | Missing value | Fresubin HP Energy | Day 1&2 | Yes | 14.0 | 26.4 | 0.80 | 1.60 |
| Patient 2 | 29.3 | 29.3 | 33.2 | Fresubin 2 kcal HP | Day 1&2 | Yes | 15.6 | 29.3 | 0.89 | 1.78 |
| Patient 3 | 40.8 | 31.3 | 37.8 | Fresubin 2 kcal HP | No | No | 19.9 | 39.9 | 1.00 | 1.99 |
| Patient 4 | 44.6 | 46.8 | Missing value | Fresubin 2 kcal HP | Day 1 | Yes | 24.9 | 44.5 | 1.38 | 2.77 |
| Patient 5 | 26.3 | 30.5 | 34.7 | Fresubin 2 kcal HP | No | No | 12.8 | 25.5 | 0.64 | 1.28 |
| Patient 6 | 29.2 | 34.5 | 38.5 | Fresubin 2kcal HP | No | No | 17.1 | 31.2 | 0.60 | 1.20 |

CRRT: Continuous renal replacement therapy; EE: Energy expenditure; EN: Enteral nutrition.

*Kcal/kg/day.

**g/kg/day.

## Discussion

The results from this randomized cross-over study suggest that 100% delivery of energy and protein targets by EN improves whole-body protein balance compared to 50%. Although the change in the primary outcome reached statistical significance, the number of patients who completed the full protocol was smaller than the target sample size due to clinical interruptions during the study period. Caution must therefore be taken in interpreting the validity of these results, and they should ideally be reproduced in a larger sample of ICU patients.

Our findings are similar to those from a previous study by our research group in neurosurgical ICU patients with 50 or 100% TPN by Berg et al [12]. While both studies are similar in design, it is important to underscore the differences between the interventions and populations studied: 1. The primary pathology of Neuro ICU patients is more homogenous which could have implications for the metabolic response to critical illness. However, the relatively high SOFA scores (mean 8) in Berg's study indicates the presence of multiorgan failure. Patients in the current study were generally later in the course of ICU stay and received limited or no sedation compared to the high dose sedation used in the Neuro ICU patients to control intracranial pressure. 2. The systemic availability of substrates from EN is more uncertain than during delivery by TPN. As successful enteral feeding was a prerequisite for inclusion in the current study, uptake by the gastrointestinal tract was most likely adequate. 3. Energy provision at full enteral feeding was not strictly "isocaloric" (caloric intake = EE), but equal to 80–

**Table 4. Whole-body phenylalanine kinetics.**

|  | 50% EN* | 100% EN* | Mean difference** | p-value† |
|---|---|---|---|---|
| Balance (μmol Phe/kg/h) | -6.1 ±1.5 | 2.9 ±2.0 | 9.0 (0.4–17.7) | 0.044 |
| Synthesis (μmol Phe/kg/h) | 56.6 ±7.9 | 65.6 ±7.3 | 9.00 (-14.2–32.2) | 0.36 |
| Breakdown (μmol Phe/kg/h) | 62.6 ±8.6 | 62.6 ±6.6 | 0.0 (-26.0–26.0) | 1.0 |
| Oxidation (μmol Phe/kg/h) | 12.6 ±2.5 | 13.7 ±2.2 | 1.10 (-8.3–10.5) | 0.78 |
| Splanchnic extraction fraction | 0.22 ±0.13 | 0.09 ±0.11 | -0.13 (-0.61–0.35) | 0.53 |
| Rate of appearance (μmol Phe/kg/h) | 70.6 ±10.0 | 80.7 ±8.7 | 10.1 (-20.1–40.2) | 0.43 |
| Plasma free amino acids (μmol/L) | 2173 ±161 | 2632 ±165 | 459 (157–762) | 0.011 |
| Serum urea (mmol/L) | 14.2 ±2.3 | 13.4 ±2.0 | -0.76 (-2.4–0.9) | 0.28 |

EN: Enteral nutrition; Phe: Phenylalanine.

* mean ± standard error.

** mean (95% confidence interval).

†All p-values calculated with two-tailed Student's T-test for paired samples.

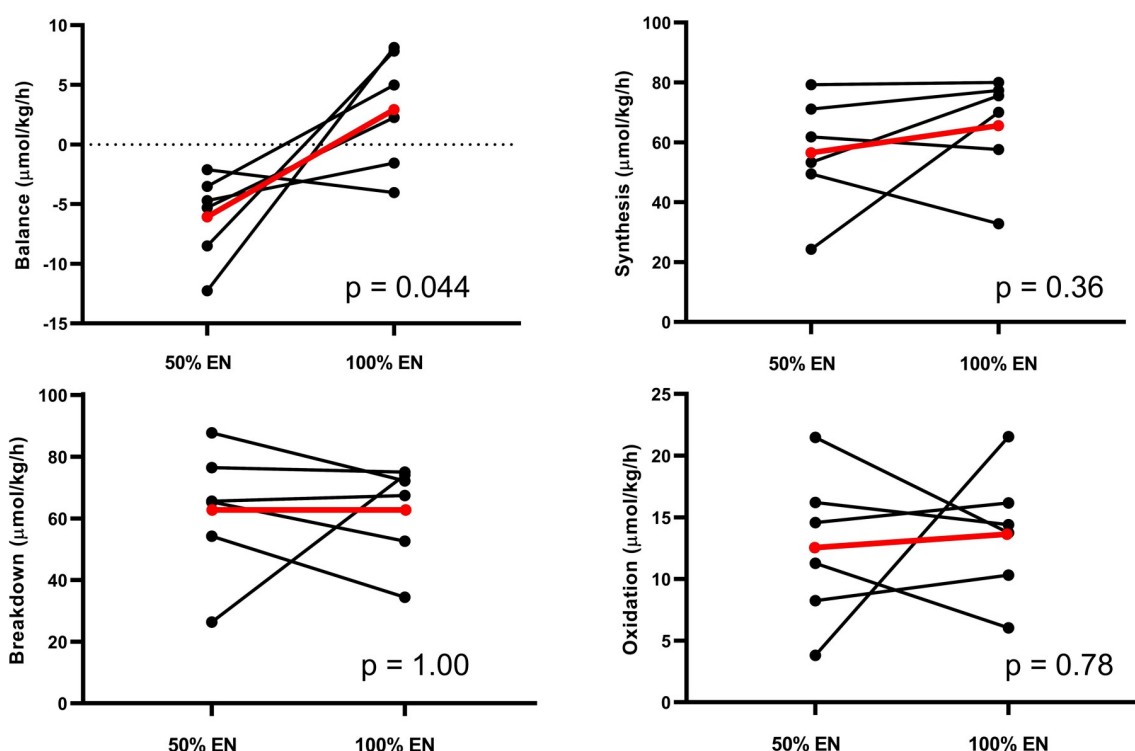

**Fig 3. Whole-body phenylalanine kinetics.** Black lines: individual patients; Red lines: mean. EN: Enteral nutrition. All p-values calculated with two-tailed Student's T-test for paired samples.

100% of measured EE. Also, protein intake was higher compared to Berg et al. (median 1.7 and 1.1 g/kg/day respectively) due to the composition of the enteral formulations used and amino acid supplementation during CRRT.

The main strength of our study is the state-of-the-art methodology used with regard to this patient population. The cross-over design allows patients to serve as their own controls, reducing the number of subjects required to observe a change in protein balance as variation in protein turnover is large between the individual critically ill patients. Previous work demonstrating improvements in protein kinetics from increased energy- or protein supplementation has primarily focused on shorter interventions in patients with total parenteral nutrition [12, 20, 21]. Here we repeat these findings in a general ICU setting during enteral feeding. Both enteral and parenteral tracers were used to allow correction for splanchnic extraction of enteral amino acids. Renal replacement therapy is a common treatment in patients with high illness severity and to our knowledge this is the first study on whole-body protein kinetics also including patients with CRRT. In these patients the loss of amino acids through the hemofilter was relatively low in comparison to whole-body turnover (<8% of total phenylalanine turnover). Furthermore, the requirement of full enteral feeding resulted in that patients were studied on days 5–30 of ICU stay. The majority were probably beyond the acute phase where mobilization of endogenous substrates may result in overfeeding if exogenous caloric support is provided to match measured EE. All together this approach allows us to perform pragmatic studies on protein turnover during critical illness without excessive constraints on inclusion criteria that may limit external validity. Although the heterogenous characteristics of patients limits inference from small samples, it is also a strength of the technique as heterogeneity is a central trait of patients in a general ICU setting. A limitation to our

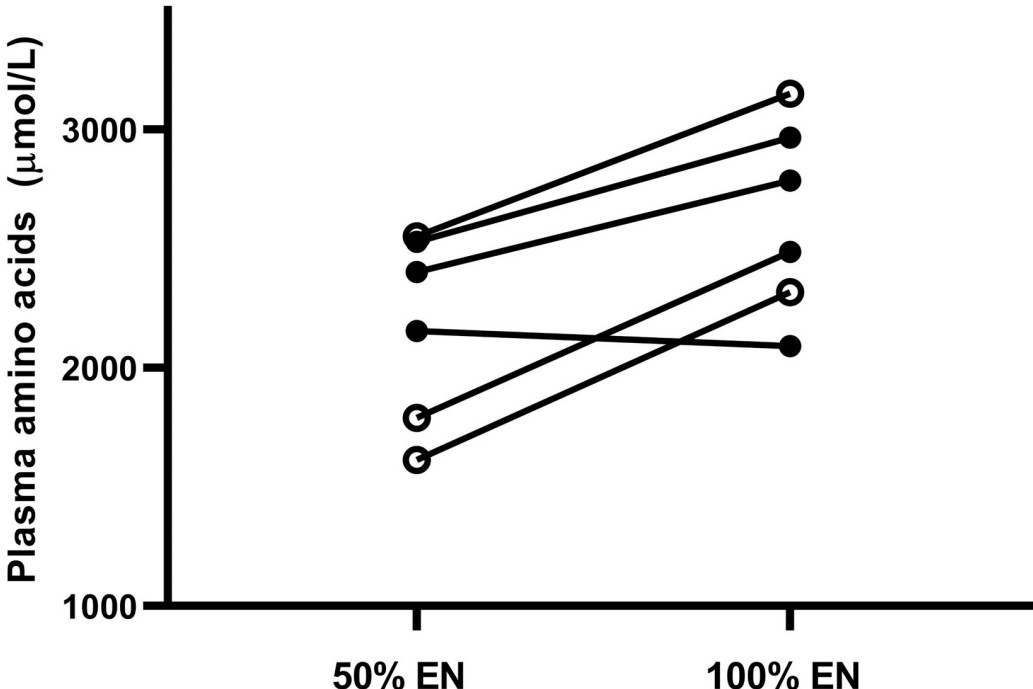

**Fig 4. Plasma free amino acid concentrations.** Black lines with full circles: Individual patients. Black lines with hollow circles: Individual patients with continuous renal replacement therapy and amino acid supplementation. EN: Enteral nutrition.

method is the long time required for adaptation to changes in nutritional therapy when using a cross-over design. The high number of dropouts due to protocol violations illustrates the difficulty in performing physiological investigations requiring longer periods of stable conditions in a modern ICU setting.

Our results indicate that critically ill patients can utilize enteral proteins to improve whole-body protein balance and that the extra proteins given during 100% compared to 50% EN are not oxidized to a larger extent. The results are also in agreement with several previous studies of amino acid kinetics critically ill patients, all demonstrating improvements in protein balance from increased energy- and/or protein supplementation [12, 15, 20–22]. However, the clinical relevance of our findings is unclear. As stated in the introduction, the available evidence from clinical trials of ICU nutrition does not support a causal link between "optimal" nutrition (as recommended in clinical practice guidelines [10, 11]) and improvements in mortality, morbidity, functional status or lean body mass. The patients enrolled in our trial illustrate two important aspects to consider in future trials of nutritional interventions; timing and individual physiologic consideration. The majority of survivors from critical illness will spend less than two weeks in the ICU [2, 23]. Nutrition does not appear to be a major determinant of short term survival, and it is unlikely that any non-vital intervention delivered for a brief period of time will have a long-term effect on physical performance status or lean body mass [24]. In our study, half of the patients had been in the ICU over three weeks, and out of those two patients exhibited a metabolic rate (measured EE at baseline) of >40 kcal/kg/day. These two patients also demonstrated the greatest increase in protein balance during full feeding (Fig 5, patient 3 and 4). Although speculative, it is physiologically plausible that patients who fail to recover become depleted in endogenous reserves and therefore are especially vulnerable to energy or protein malnutrition. This is potentially compounded by unrecognized hypermetabolism in

**Table 5. Plasma amino acid profiles of individual patients.**

| | %EN | 3-MH | Ala | Arg | Asn | Cit | Gln | Glu | Gly | His | Ile | Leu | Lys | Met | Orn | Phe | Ser | Tau | Thr | Trp | Tyr | Val | EAA | BCAA | SUM |
|---|---|---|---|---|---|---|---|---|---|---|---|---|---|---|---|---|---|---|---|---|---|---|---|---|---|
| Patient 1 | 50 | 8.1 | 138.9 | 55.3 | 36.5 | 21.7 | 308.8 | 73.7 | 121.1 | 58.8 | 46.6 | 83.7 | 130.9 | 17.5 | 61.0 | 61.4 | 56.2 | 33.2 | 69.7 | 27.5 | 62.1 | 140.2 | 636.3 | 270.5 | 1612.8 |
| Patient 2 | 50 | 7.2 | 256.4 | 50.6 | 47.0 | 52.3 | 644.8 | 93.4 | 216.3 | 64.6 | 79.8 | 132.0 | 129.5 | 18.0 | 114.1 | 108.5 | 75.7 | 37.5 | 84.5 | 33.1 | 75.1 | 229.9 | 879.9 | 441.8 | 2550.4 |
| Patient 3 | 50 | 5.3 | 168.9 | 65.7 | 40.0 | 34.3 | 602.7 | 157.2 | 156.8 | 41.4 | 60.4 | 105.9 | 142.3 | 16.6 | 154.0 | 86.3 | 72.3 | 183.6 | 88.3 | 22.7 | 73.9 | 248.9 | 812.8 | 415.2 | 2527.5 |
| Patient 4 | 50 | 4.1 | 215.0 | 28.1 | 25.2 | 1.4 | 379.7 | 148.0 | 85.6 | 37.5 | 51.6 | 115.1 | 56.5 | 8.3 | 63.3 | 124.4 | 41.2 | 97.3 | 30.3 | 22.0 | 48.3 | 207.5 | 653.1 | 374.3 | 1790.4 |
| Patient 5 | 50 | 27.5 | 187.8 | 78.2 | 46.7 | 50.1 | 533.5 | 58.1 | 210.2 | 50.3 | 56.5 | 96.9 | 157.8 | 26.6 | 75.3 | 238.1 | 56.0 | 46.3 | 111.7 | 29.1 | 71.4 | 193.2 | 960.2 | 346.5 | 2401.1 |
| Patient 6 | 50 | 6.7 | 161.0 | 54.5 | 42.3 | 41.0 | 539.6 | 23.3 | 191.3 | 48.7 | 54.8 | 113.5 | 173.9 | 26.3 | 69.9 | 96.0 | 70.2 | 33.2 | 116.0 | 17.2 | 60.4 | 214.8 | 861.1 | 383.0 | 2154.3 |
| | %EN | 3-MH | Ala | Arg | Asn | Cit | Gln | Glu | Gly | His | Ile | Leu | Lys | Met | Orn | Phe | Ser | Tau | Thr | Trp | Tyr | Val | EAA | BCAA | SUM |
| Patient 1 | 100 | 7.1 | 181.8 | 72.1 | 46.4 | 27.6 | 509.1 | 86.8 | 147.1 | 74.3 | 69.0 | 132.0 | 183.4 | 22.2 | 79.6 | 90.2 | 81.1 | 32.7 | 110.2 | 34.4 | 102.1 | 228.2 | 943.9 | 429.3 | 2317.1 |
| Patient 2 | 100 | 6.0 | 445.6 | 59.6 | 59.3 | 50.7 | 790.9 | 81.4 | 239.5 | 76.6 | 74.7 | 139.2 | 148.9 | 24.9 | 127.7 | 138.5 | 81.1 | 80.3 | 114.2 | 34.0 | 110.6 | 265.7 | 1016.8 | 479.5 | 3149.4 |
| Patient 3 | 100 | 5.3 | 200.9 | 80.9 | 47.4 | 37.7 | 632.0 | 160.0 | 166.2 | 47.3 | 76.0 | 146.8 | 201.0 | 24.3 | 159.3 | 120.8 | 78.8 | 182.5 | 134.2 | 27.7 | 105.1 | 331.7 | 1109.7 | 554.5 | 2966.1 |
| Patient 4 | 100 | 5.8 | 331.0 | 33.4 | 38.2 | 36.9 | 608.8 | 155.7 | 122.9 | 45.3 | 95.4 | 159.8 | 84.7 | 15.3 | 84.1 | 144.9 | 52.9 | 38.7 | 48.6 | 28.4 | 74.8 | 279.6 | 902.2 | 534.8 | 2485.1 |
| Patient 5 | 100 | 24.9 | 240.5 | 74.4 | 55.5 | 52.1 | 613.6 | 77.1 | 239.6 | 59.0 | 68.9 | 119.7 | 150.5 | 31.2 | 97.9 | 264.9 | 56.2 | 39.4 | 123.1 | 39.2 | 85.0 | 272.1 | 1128.6 | 460.7 | 2784.8 |
| Patient 6 | 100 | 7.1 | 133.4 | 46.6 | 42.9 | 32.8 | 539.4 | 27.0 | 183.6 | 54.0 | 46.7 | 96.8 | 175.3 | 22.7 | 64.5 | 92.9 | 63.7 | 44.7 | 137.7 | 15.3 | 55.3 | 208.6 | 850.0 | 352.1 | 2090.9 |

Plasma amino acid concentrations are given as μmol/L. Individual amino acids are presented with standard abbreviations. BCAA: Branched chain amino acids; EAA: Essential amino acids; EN: Enteral nutrition; SUM: Sum of all measured amino acids.

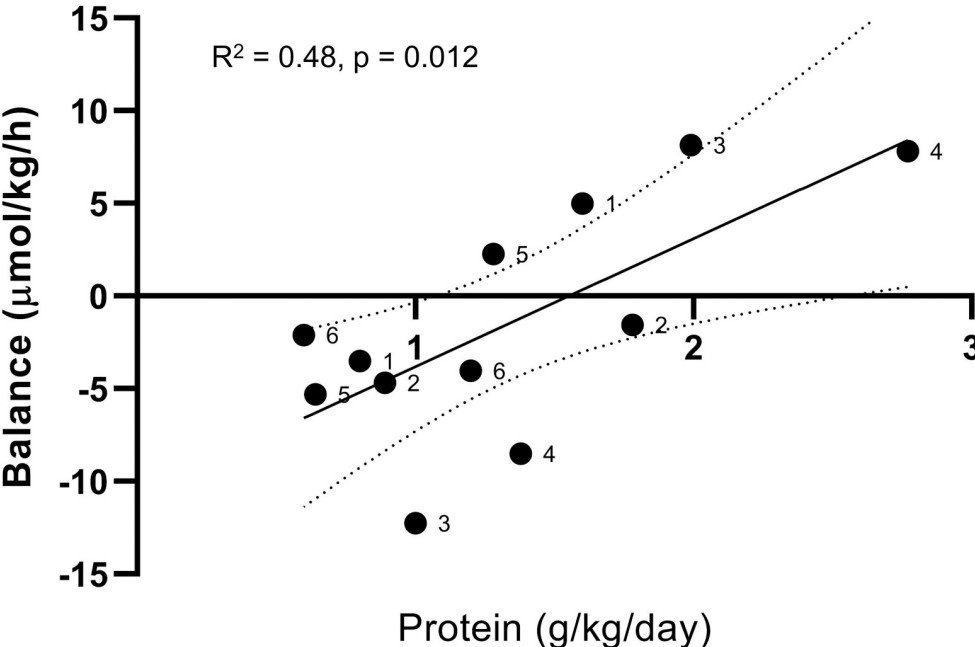

**Fig 5. Whole-body phenylalanine balance in relation to protein intake.** Black dots: individual patients with assigned numbers. Solid black line: Simple linear regression of data points. Dotted lines: 95% confidence interval of regression line.

the absence of indirect calorimetry. As this subgroup only represent a small fraction of patients included in major RCTs, very limited inference can be drawn from available evidence. It is imperative for future research to better define the role of individualized nutritional therapy in ICU patients who suffer from the nebulous condition of chronic critical illness. Measurements of protein kinetics provide unique information on treatment response and can be used in conjunction with other modalities that measure change in lean body mass or functional status [25].

Our study has several weaknesses, mainly a smaller-than-planned per protocol sample size that limits the strength of our findings. Second, the heterogeneity of our sample with regards to age, sex, disease state and time spent in the ICU limits interpretations of our results to hypothesis-generating conclusions. Third, the protein-to-calorie ratio of administered nutritional formulas was kept constant throughout the study. We are therefore unable to discern whether the effects on protein kinetics are due to increased energy provision, exogenous protein supply or both. Fourth, whole-body protein balance does not give any indication as to which organ systems experience a nitrogen sparing effect or if these changes are relevant to recovery. Fifth, we cannot determine if the observed changes are attenuated by adaptation over time beyond the frame of the current study [26].

## Conclusions

This study demonstrates that a stable isotope tracer technique applied in a cross-over design allows the evaluation of how nutrition affects protein metabolism in critically ill patients during enteral feeding and CRRT. Our results suggest that 100% of prescribed EN improves whole-body protein balance compared to 50% of EN in a group of patients with known energy expenditure and established enteral feeding, but the validity of this finding needs to be confirmed in a larger investigation. Future studies using a similar methodology in a larger sample

could provide relevant information to guide nutritional therapy during long-term critical illness.

## Supporting information

**S1 Checklist. CONSORT 2010 checklist.**
(DOCX)

**S1 File. Ethical application and study protocol (original language).**
(PDF)

**S2 File. Study protocol from ethical application (English translation).**
(DOCX)

## Acknowledgments

Our most heartfelt thanks to our invaluable co-workers: Kristina Kilsand, Sara Rydén and Janelle Cederlund for performing the clinical studies, Eva Nejman, Towe Jakobsson and Christina Hebert for performing the laboratory analyses and Daniel Olsson for statistical advice.

## Author Contributions

**Conceptualization:** Felix Liebau, Jan Wernerman, Olav Rooyackers.

**Data curation:** Martin Sundström Rehal, Olav Rooyackers.

**Formal analysis:** Martin Sundström Rehal, Olav Rooyackers.

**Funding acquisition:** Jan Wernerman.

**Investigation:** Olav Rooyackers.

**Methodology:** Felix Liebau, Jan Wernerman, Olav Rooyackers.

**Project administration:** Martin Sundström Rehal, Felix Liebau, Jan Wernerman, Olav Rooyackers.

**Resources:** Olav Rooyackers.

**Supervision:** Jan Wernerman, Olav Rooyackers.

**Visualization:** Martin Sundström Rehal.

**Writing – original draft:** Martin Sundström Rehal.

**Writing – review & editing:** Martin Sundström Rehal, Felix Liebau, Jan Wernerman, Olav Rooyackers.

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
