## [Decision Letter · Decision Letter 0]

4 May 2020

PONE-D-20-06211

Whole body protein kinetics in critically ill patients during 50 or 100% energy provision by enteral nutrition: A pilot randomized cross-over study

PLOS ONE

Dear Dr Sundström Rehal,

Thank you for submitting your manuscript to PLOS ONE. After careful consideration, we feel that it has merit but does not fully meet PLOS ONE’s publication criteria as it currently stands. Therefore, we invite you to submit a revised version of the manuscript that addresses the points raised during the review process.

We would appreciate receiving your revised manuscript by Jun 18 2020 11:59PM. To enhance the reproducibility of your results, we recommend that if applicable you deposit your laboratory protocols in protocols.io, where a protocol can be assigned its own identifier (DOI) such that it can be cited independently in the future. For instructions see: http://journals.plos.org/plosone/s/submission-guidelines#loc-laboratory-protocols

We look forward to receiving your revised manuscript.

Kind regards,

Juan J Loor

Academic Editor

PLOS ONE

Journal Requirements:

"I have read the journal's policy and the authors of this manuscript have the following

competing interests: JW and OR have given paid lectures about nutrition in the ICU for

Nestlé, Nutricia and Fresenius Kabi. OR is a consultant for Fresenius-Kabi. FL has

received a speaking fee from Baxter. MSR has no competing interests to declare."

Reviewers' comments:

Reviewer's Responses to Questions

**Comments to the Author**

1. Is the manuscript technically sound, and do the data support the conclusions?

Reviewer #1: Yes

Reviewer #2: Yes

Reviewer #3: Partly

2. Has the statistical analysis been performed appropriately and rigorously? 

Reviewer #1: Yes

Reviewer #2: Yes

Reviewer #3: No

3. Have the authors made all data underlying the findings in their manuscript fully available?

Reviewer #1: No

Reviewer #2: Yes

Reviewer #3: No

4. Is the manuscript presented in an intelligible fashion and written in standard English?

Reviewer #1: Yes

Reviewer #2: Yes

Reviewer #3: Yes

5. Review Comments to the Author

Reviewer #1: Overall the study appears to have been well conducted and the limitations are clearly stated. the writing is clear and concise. I only have a few minor comments.

1) The ages of the participants should be included in table 1

2) the trial registration states that Splanchnic extraction is a secondary outcome however these data are not included in the paper, instead the they appear to be used to correct protein balance data. Why is Splanchnic extraction not reported?

3) The secondary outcome listed in the introduction is "plasma amino acid profile" however the only related result reported is the total concentration of all amino acids summed together. it is not clear how these samples were analyzed and why concentrations of each AA are not reported to generate a profile

4) the laboratory analytic methods (mass spec, ect) not reported in sufficient detail to be replicated. No references are provided which show how blood samples (serum, plasma, EDTA, heparin?) were processed what what instruments they were analyzed on.

5) the timeline figure is very hard to follow. I suggest a timeline running along the center of the figure with rectangles to represent the infusion periods

6) please indicate statistical significance directly on figures where appropriate

7) the raw data on which the paper is based should be uploaded as a supplemental file

Reviewer #2: The workers targeted a high-risk study population, and their efforts to help improve the clinical outcomes of ICU patients is certainly commendable. However, there are some concerns about the small and heterogenous sample population, confounding and inconsistent inclusion of CRRT, and overall practicality of outcomes application in the clinical setting lead to some reservations on the validity of this research in its current state. I recognize this is pilot work, but perhaps the messaging would be more useful if the authors continued to recruit patients to achieve a full data set for the a priori determined sample size and make efforts to reduce patient variability (e.g., CRRT, disease severity, length of stay, etc) before deducing clinical procedures from their results.

Major Comments

• Why include CRRT? The impact of dialysis/renal failure (even if acute) on protein metabolism and amino acid availability are highly confounding to include this pathology and treatment with non-CRRT patients. Further, it is also noted that one of the CRRT patients only received CRRT on 1 out of 2 of the experimental crossover days (Table 3). This precludes the ability of that patient to serve as their own control.

• Final full-data set subject count (n=6) is well below a priori power calculation (n=10). Even then, of the 6, some variables have missing data from participants. While p<0.05 was observed in protein balance and plasma amino acids, the other variables remain highly variable.

• For patients on CRRT during the infusion, the authors only mention accounting for amino acid concentrations. However, what about tracer removal, and subsequent impact on blood enrichment? Was this accounted for in calculations. Please articulate.

• It is indicated that IV tracer infusion was stopped to perform IC in between experimental conditions. Was the stop unanimous between patients? How does this influence tracer kinetics thus calculations?

• Trial 2 isotopic infusion initiates only 19 h after stopping previous IV infusion. How does presence of isotope from previous infusion influence trial 2 outcomes? Specifically, 13C-Phe within continuous EN feed from previous experiment impacting the 2nd experiment (no stop of 13C-Phe enriched EN). Also, the bolus of D4-Tyr: is enrichment of the previous bolus still detectable? How does this influence calculation of current turnover kinetics?

• There is considerable variation in SAPS II, SOFA and LoS between patients. These variables certainly influence protein metabolism, thus homogeneity of sample population for outcome measures.

• All tables/figures are not stand alone. These should be inclusive for descriptive statistics and statistical comparisons.

• It would be interesting to note how many potential participants were excluded due to not meeting inclusion of >= 80% EE needs by IC. These data will provide insight on how practical the current effort is for clinical application. Furthermore, 1 out 6 feeding intolerance was noted. Even if protein balance may be improved based on these results from a heterogenous and insufficient sample size, how achievable is 100% goal rate to achieve >80% EEN in the ICU setting?

Minor Comments

• Line 83-84 indicates revised inclusion/exclusion criteria. It is unclear if the previously mentioned criteria were the original or updated. This should be better articulated

• Study protocol figure is confusing and not clear.

• Line 119: The ring-D4-Tyr bolus is not clarified as within EN or IV.

• What is the justification for performing IC so frequently (baseline inclusion, between crossover, and end of trial 2)? Is this really necessary? Especially considering IC is not widely available or used in the clinical setting.

• Inconsistency in units for infusion rate within the manuscript: Lines 118-119: mg/kg/h versus Line 146: umol/kg/min.

• Similarly, inconsistent units for outcome between power calculation (Line 162: mg Phe/kg/h) versus Table 4 results (umol Phe/kg/h).

• Table 4: What is the calculation for serum/plasma variables? Is this weighted average? Area under the curve? Please specifiy.

• Table 4: Multiple blood samples were collected, but only mean cumulative values are reported. Response curves and appropriate statistical tests may provide valuable information.

• Unclear why expressing isotopic enrichments as MPE? TTR is more appropriate. Graphs showing the isotopic enrichments would be helpful to interpret your results and modeling.

Reviewer #3: The authors report a small (n=6) trial investigating 50% vs. 100% energy provision via enteric nutrition in ICU patients for the effect on protein balance, as measured by the kinetics of phenylalanine. My decision to recommend rejection of this manuscript is based on the fact that while the title of the study suggests that it is a feasibility study, there is nothing in the study report which shows that the authors were investigating the feasibility of investigating this intervention in a full trial. None of the usual outcomes of a feasibility/pilot trial (e.g. a timeframe for recruiting and randomising a certain number of patients which would demonstrate feasibility to scale up to a full trial) are even mentioned, and it is impossible to determine how the authors arrive at the conclusion that: "It is feasible to assess whole body protein turnover using a stable isotope technique in critically ill patients during enteral feeding and renal replacement therapy".

The manuscript is presented as if this was a full trial. If judged on this criterion, the study is way too small to reasonable assess the role of chance in the observed differences. The small size leaves the authors no scope to consider balancing the sample for baseline characteristics that could influence the outcome. Indeed there's no exploration at all of such an important issue. The size of the study is poorly justified, not reported in a standard format, and the calculation is not reproducible.

I would recommend that the authors offer clarity on whether this was in fact a feasibility study, and if so focus the reporting on demonstrating feasibility to scale it up to a full trial, with the outcomes reported here considered secondary in such a report, or whether this was really it - a small trial - and if so to report it and present all the caveats that would be necessary in the interpretation of such a study.

6. PLOS authors have the option to publish the peer review history of their article (what does this mean?). If published, this will include your full peer review and any attached files.

Reviewer #1: No

Reviewer #2: No

Reviewer #3: No

---

## [Author Response · Author response to Decision Letter 0]

7 Jul 2020

The authors would like to thank the academic editor and all reviewers for providing a very thorough and thoughtful review of our manuscript. Many important points have been raised which we have attempted to address below. In the instances where revisions have been made to the manuscript the line numbers refer to the marked-up copy. 

Response to comments from the editorial office

To enhance the reproducibility of your results, we recommend that if applicable you deposit your laboratory protocols in protocols.io, where a protocol can be assigned its own identifier (DOI) such that it can be cited independently in the future. For instructions see: http://journals.plos.org/plosone/s/submission-guidelines#loc-laboratory-protocols

We agree that this is sound practice. Many of our laboratory protocols are written in Swedish and due to the extra workload from the COVID-19 pandemic the authors have not been able to translate these documents within the time frame for review of this manuscript. We will however attempt to achieve this for future purposes. 

We have gone over the manuscript again, editing tables and file names in accordance with PLOS ONE’s requirements. This should be in order. 

2. We note that you have indicated that data from this study are available upon request. PLOS only allows data to be available upon request if there are legal or ethical restrictions on sharing data publicly. 

The majority of all data collected for the study is openly available at the Swedish National Data Service database (https://doi.org/10.5878/b1e8-fg58). The only restrictions we have indicated concern open access to patient characteristics, as the small number of patients in this study and time frame for data collection provided may facilitate unintentional de-identification of patients. The ethical permit for our study is only valid under conditions of preserved patient anonymity. 

"I have read the journal's policy and the authors of this manuscript have the following

competing interests: JW and OR have given paid lectures about nutrition in the ICU for

Nestlé, Nutricia and Fresenius Kabi. OR is a consultant for Fresenius-Kabi. FL has

received a speaking fee from Baxter. MSR has no competing interests to declare."

The competing interests section has been updated accordingly.

Captions for the supporting information files have been included at the end of the manuscript. 

Response to comments from reviewers

Reviewer #1: Overall the study appears to have been well conducted and the limitations are clearly stated. the writing is clear and concise. I only have a few minor comments. 

1) The ages of the participants should be included in table 1

We agree that this is relevant information. Due to the small number of patients included, this was omitted from the manuscript to avoid unintentional identification of patients. After careful consideration we have included this information to Table 1. as the admission diagnosis and age alone cannot be connected to a specific patient i.d. by a third party with any degree of certainty. 

2) the trial registration states that Splanchnic extraction is a secondary outcome however these data are not included in the paper, instead the they appear to be used to correct protein balance data. Why is Splanchnic extraction not reported?

This was an oversight. Although not of interest as an outcome measure on its own, estimating the splanchnic extraction ratio of enteral phenylalanine is a necessary intermediate step in the calculation of whole-body protein balance. Splanchnic extraction fraction is now reported in Table 4. and the section “Calculations” (Line 143) has been expanded to clarify how the full balance calculations were performed. Some numbers are negative, which would imply that the splanchnic organs are a net contributor of phenylalanine beyond exogenous amino acid uptake. From our data we cannot deduce if this is an accurate physiological observation or if these numbers only represent variability around net zero. 

3) The secondary outcome listed in the introduction is "plasma amino acid profile" however the only related result reported is the total concentration of all amino acids summed together. it is not clear how these samples were analyzed and why concentrations of each AA are not reported to generate a profile

We agree that this information should be reported and have added amino acid profiles for individual patients in Table 5. Information about analytical methods for the quantification of plasma amino acid concentrations have been added to the “Methods” section (line 143-144). 

4) the laboratory analytic methods (mass spec, ect) not reported in sufficient detail to be replicated. No references are provided which show how blood samples (serum, plasma, EDTA, heparin?) were processed what what instruments they were analyzed on.

We have expanded the reporting of this information, provided under “Methods/Protocol” (line 137-145). 

5) the timeline figure is very hard to follow. I suggest a timeline running along the center of the figure with rectangles to represent the infusion periods

We agree that Figure 2. needs be clarified and have attempted to incorporate your suggestions in a revised figure. 

6) please indicate statistical significance directly on figures where appropriate

Table 4, Figure 3 and Figure 5 have been updated according to this suggestion. 

7) the raw data on which the paper is based should be uploaded as a supplemental file

In accordance with PLOS ONEs editorial policy we have uploaded the raw data in an open repository (https://doi.org/10.5878/b1e8-fg58)

Reviewer #2: The workers targeted a high-risk study population, and their efforts to help improve the clinical outcomes of ICU patients is certainly commendable. However, there are some concerns about the small and heterogenous sample population, confounding and inconsistent inclusion of CRRT, and overall practicality of outcomes application in the clinical setting lead to some reservations on the validity of this research in its current state. I recognize this is pilot work, but perhaps the messaging would be more useful if the authors continued to recruit patients to achieve a full data set for the a priori determined sample size and make efforts to reduce patient variability (e.g., CRRT, disease severity, length of stay, etc) before deducing clinical procedures from their results.

We fully understand the concerns about the small number of patients included and problems with heterogeneity in this study. Hopefully we can provide better context regarding sample size and patient variability through the answers below:

1. We agree that the preferable course of action would have been to continue recruitment until completing inclusion of the intended number of patients with data for the primary outcome. Our ability to proceed in this direction was restricted by unfortunate external events. Due to a restructuring of the regional pharmaceutical organization in Stockholm County, we lost our supplier of tracer preparations licensed for clinical use. As we did not see a solution to this issue in the foreseeable future, we decided to submit our work in its current form as we believe the method described is unique and may be of value to future research. It definitely limits the ability to draw inference regarding the primary outcome (protein balance). Our intention was to reflect this in the “Discussion” and “Conclusions” section of the manuscript. These have been revised to further clarify the limitations/scope of our study. 

2. The inclusion/exclusion criteria of this study (established full enteral nutrition, FiO2 <0.6, no imminent extubation/patient transfer during the study period) tend to “favor” patients who are in a stable, prolonged state of their ICU course and therefore diverge in clinical characteristics. Despite conducting recruitment over 15 months with dedicated research nurses screening for potential subjects, only 12 patients were included. Narrowing our inclusion criteria to reduce heterogeneity would have made recruitment even more difficult. Although heterogeneity and small sample size limits the external validity of our results, we consider the successful application of our method in a heterogenous ICU population a strength of our study. Also, we believe that the cross-over design reduces the effects of heterogeneity in interpreting changes in the primary outcome. 

3. We want to be very clear that we are not deducing any clinical procedures from our results. We believe that our data shows a signal towards an improved whole body protein balance from full enteral feeding. Interpretations are limited by the small sample size, and we do not consider changes in protein balance a patient-centered outcome on which to base clinical recommendations. However, we believe that our results and the methods used have merits to understanding the physiology of metabolism in ICU patients, and that they are important primarily for future research. 

Major Comments

• Why include CRRT? The impact of dialysis/renal failure (even if acute) on protein metabolism and amino acid availability are highly confounding to include this pathology and treatment with non-CRRT patients. Further, it is also noted that one of the CRRT patients only received CRRT on 1 out of 2 of the experimental crossover days (Table 3). This precludes the ability of that patient to serve as their own control.

The authors respectfully disagree with the reviewer in this case. Patients with acute kidney injury represent a subset of ICU patients with higher illness severity & risk of death, longer durations of ICU stay and increased risk of muscle loss and long-term functional disability. Developing research methods to better understand the metabolic alterations in these patients is of particular importance. It is our opinion that the inclusion of patients with CRRT is a strength & novelty of our study. We believe the method applied is robust and useful for future research. The “Calculations” section under “Methods” has been revised to provide more detail how protein balance is corrected for loss of amino acids over the hemofilter. 

Regarding the patient who only received CRRT during one of the study periods, this does not preclude that the patient can serve as its own control as the change in protein balance is corrected for a known loss (phenylalanine concentration in the effluent). In this study phenylalanine loss via CRRT only had a minor effect (<8%) on the calculation of whole-body protein balance. Other factors (variable splanchnic extraction or uptake of enteral amino acids, protein oxidation etc) are potentially more significant sources of variability between study periods, which the method also attempts to correct for. 

• Final full-data set subject count (n=6) is well below a priori power calculation (n=10). Even then, of the 6, some variables have missing data from participants. While p<0.05 was observed in protein balance and plasma amino acids, the other variables remain highly variable.

We agree that the final number of subjects is lower than our original intentions, for reasons described above. P-values and interpretations of our results should therefore be interpreted with caution. The “Discussions” section of the manuscript has been revised to state this with further emphasis. 

• For patients on CRRT during the infusion, the authors only mention accounting for amino acid concentrations. However, what about tracer removal, and subsequent impact on blood enrichment? Was this accounted for in calculations. Please articulate.

The method assumes that, at steady state, Phenylalanine and D5-Phenylalanine are removed from the central compartment at an equal rate. As the rate of appearance (Ra) is calculated from the dilution of tracer at the enrichment plateau, loss of tracer over the hemofilter will not affect Ra as it is not part of any calculation. Quantifying the rate of tracee (phenylalanine) loss to dialysis is important to distinguish this from synthesis and oxidation as a contribution to rate of disappearance (Rd). 

• It is indicated that IV tracer infusion was stopped to perform IC in between experimental conditions. Was the stop unanimous between patients? How does this influence tracer kinetics thus calculations?

Enteral and parenteral tracers are only infused during the last five hours of the 24 hour intervention/control period. After the first period is complete (after blood sampling), no further tracers are administered until the end (final 5 hours) of the next 24 hour period. Figure 2. has been revised for greater clarity. 

• Trial 2 isotopic infusion initiates only 19 h after stopping previous IV infusion. How does presence of isotope from previous infusion influence trial 2 outcomes? Specifically, 13C-Phe within continuous EN feed from previous experiment impacting the 2nd experiment (no stop of 13C-Phe enriched EN). Also, the bolus of D4-Tyr: is enrichment of the previous bolus still detectable? How does this influence calculation of current turnover kinetics?

13C-Phe was not administered by directly supplementing the enteral formula, it was provided as a separate infusion connected to the patient's nasogastric line by a three-way stopcock. At the end of the first measurement period this infusion was stopped and standard EN formula continued at the new rate. Arterial enrichments of all tracers were back to baseline at the start of the second measurement period. The “Protocol” section has been revised for greater clarity. 

• There is considerable variation in SAPS II, SOFA and LoS between patients. These variables certainly influence protein metabolism, thus homogeneity of sample population for outcome measures.

We agree that this limits the interpretation of our results. 

• All tables/figures are not stand alone. These should be inclusive for descriptive statistics and statistical comparisons.

We have revised all figures/legends according to your recommendations. 

• It would be interesting to note how many potential participants were excluded due to not meeting inclusion of >= 80% EE needs by IC. These data will provide insight on how practical the current effort is for clinical application. Furthermore, 1 out 6 feeding intolerance was noted. Even if protein balance may be improved based on these results from a heterogenous and insufficient sample size, how achievable is 100% goal rate to achieve >80% EEN in the ICU setting?

Our opinion (outside the scope of this study) is that the majority of our patients meet their caloric targets by enteral nutrition after the first week of ICU stay. The CONSORT flowchart has been updated with reasons for exclusion to show how common “inadequate” enteral nutrition was in all cases initially screened. 

Minor Comments

• Line 83-84 indicates revised inclusion/exclusion criteria. It is unclear if the previously mentioned criteria were the original or updated. This should be better articulated

This has been clarified in the revised manuscript, Line 84-85. 

• Study protocol figure is confusing and not clear.

We agree and have revised the figure for greater clarity. 

• Line 119: The ring-D4-Tyr bolus is not clarified as within EN or IV.

I.V., this is now stated clearly in the manuscript on Line 120. 

• What is the justification for performing IC so frequently (baseline inclusion, between crossover, and end of trial 2)? Is this really necessary? Especially considering IC is not widely available or used in the clinical setting.

The purpose of performing multiple indirect calorimetries is to ascertain that the two intervention periods are comparable from a metabolic research perspective. Unfortunately it was not possible to measure EE in all patients according to protocol. We do not advocate for measurements this frequently for clinical purposes. 

• Inconsistency in units for infusion rate within the manuscript: Lines 118-119: mg/kg/h versus Line 146: umol/kg/min.

The manuscript has been double-checked for inconsistencies regarding units and all instances should be corrected in the revised manuscript and given as umol/kg/h.

• Similarly, inconsistent units for outcome between power calculation (Line 162: mg Phe/kg/h) versus Table 4 results (umol Phe/kg/h).

• Table 4: What is the calculation for serum/plasma variables? Is this weighted average? Area under the curve? Please specifiy.

Kinetic data is based on average values from the four blood samples drawn at the end of the tracer period. Amino acid and urea values are single point measurements. This has been clarified in Line 137-145 of the manuscript. 

• Table 4: Multiple blood samples were collected, but only mean cumulative values are reported. Response curves and appropriate statistical tests may provide valuable information.

Performing tracer kinetics using averages of 4 samples is standard practice in tracer studies. Raw data for calculation of plasma enrichment is available in the source data repository. 

• Unclear why expressing isotopic enrichments as MPE? TTR is more appropriate. Graphs showing the isotopic enrichments would be helpful to interpret your results and modeling.

We know that both terms are used but we find MPE more appropriate for stable isotopes because it takes the actual mass contribution of the tracer into consideration. The calculations for TTR and MPE are slightly different and when the right formulas are used the results are the same. The mass spectrometry measurements from blood samples used to calculate isotopic enrichment are available in the open data repository. We have chosen not to include visual depictions of isotopic enrichments in the manuscript as this would necessitate a minimum of 24 individual graphs for D5/13C-phenylalanine. 

Reviewer #3: The authors report a small (n=6) trial investigating 50% vs. 100% energy provision via enteric nutrition in ICU patients for the effect on protein balance, as measured by the kinetics of phenylalanine. My decision to recommend rejection of this manuscript is based on the fact that while the title of the study suggests that it is a feasibility study, there is nothing in the study report which shows that the authors were investigating the feasibility of investigating this intervention in a full trial. None of the usual outcomes of a feasibility/pilot trial (e.g. a timeframe for recruiting and randomising a certain number of patients which would demonstrate feasibility to scale up to a full trial) are even mentioned, and it is impossible to determine how the authors arrive at the conclusion that: "It is feasible to assess whole body protein turnover using a stable isotope technique in critically ill patients during enteral feeding and renal replacement therapy".

The manuscript is presented as if this was a full trial. If judged on this criterion, the study is way too small to reasonable assess the role of chance in the observed differences. The small size leaves the authors no scope to consider balancing the sample for baseline characteristics that could influence the outcome. Indeed there's no exploration at all of such an important issue. The size of the study is poorly justified, not reported in a standard format, and the calculation is not reproducible.

I would recommend that the authors offer clarity on whether this was in fact a feasibility study, and if so focus the reporting on demonstrating feasibility to scale it up to a full trial, with the outcomes reported here considered secondary in such a report, or whether this was really it - a small trial - and if so to report it and present all the caveats that would be necessary in the interpretation of such a study.

We want to thank Reviewer #3 for illuminating several important limitations of our study. However, there are certain points which we wish to clarify.

1. We understand that our use of the word “feasible” in the conclusions may cause confusion about the aims and scope of this study. To be clear, this is not a feasibility study and it was never our intention to report it as such. The reviewer correctly identifies it as a small clinical trial. Due to the limitations discussed below (and in the response to comments from Reviewers #1 & 2) we want to be restrictive in the interpretation of our results regarding protein kinetics during critical illness. The term “feasible” was used as we believe that the method applied is novel in critically ill patients, could be performed in a diverse group of subjects, is useful for future research, and that this is a conclusion we can stand by. To avoid other interpretations the manuscript has been rephrased. 

2. We agree that heterogeneity in baseline characteristics limits interpretations of our results. This will always be the case in small studies of critically ill patients as the only common denominator of these patients is a severity of illness that warrants ICU admission. Narrowing the inclusion criteria to decrease heterogeneity would raise other questions about the method’s utility in a more diverse sample of patients (and also greatly increase difficulties in recruitment). To balance this we chose to use patients as their own controls, as this reduces the impact of between-group differences in characteristics when interpreting results. 

3. The small size also limits interpretation of our results. The manuscript has been revised to clarify the sample size calculation. Unfortunately, for reasons previously described in our reply to Reviewer #2, recruitment fell short of the original target. As Reviewer #3 correctly points out, the results presented may be attributable to random effects in a small group of patients, although they are consistent with earlier observations in other studies. It was our intention to communicate this degree of uncertainty in the original manuscript. The Discussion/Conclusions sections have been revised to further stress these limitations.

---

## [Decision Letter · Decision Letter 1]

17 Jul 2020

PONE-D-20-06211R1

Whole body protein kinetics in critically ill patients during 50 or 100% energy provision by enteral nutrition: A pilot randomized cross-over study

PLOS ONE

Dear Dr. Sundström Rehal,

Thank you for submitting your manuscript to PLOS ONE. After careful consideration, we feel that it has merit but does not fully meet PLOS ONE’s publication criteria as it currently stands. Therefore, we invite you to submit a revised version of the manuscript that addresses the points raised during the review process.

We look forward to receiving your revised manuscript.

Kind regards,

Juan J Loor

Academic Editor

PLOS ONE

Reviewers' comments:

Reviewer's Responses to Questions

**Comments to the Author**

1. If the authors have adequately addressed your comments raised in a previous round of review and you feel that this manuscript is now acceptable for publication, you may indicate that here to bypass the “Comments to the Author” section, enter your conflict of interest statement in the “Confidential to Editor” section, and submit your "Accept" recommendation.

Reviewer #3: (No Response)

2. Is the manuscript technically sound, and do the data support the conclusions?

Reviewer #3: Partly

3. Has the statistical analysis been performed appropriately and rigorously? 

Reviewer #3: No

4. Have the authors made all data underlying the findings in their manuscript fully available?

Reviewer #3: Yes

5. Is the manuscript presented in an intelligible fashion and written in standard English?

Reviewer #3: Yes

6. Review Comments to the Author

Reviewer #3: The authors have made a commendable effort to respond to previous reviewers' comments. Although I agree that this is an important and novel study, I am still not satisfied with the responses to my previous comments relating to whether this is a pilot study or a small trial.

"Some recent reviews ... highlighted that sometimes a small underpowered effectiveness study is labelled as a pilot or feasibility study. There is therefore a need to raise awareness of the difference between a pilot study which is designed to clarify areas of uncertainty, and a small underpowered study labelled as a pilot which does not comply with definitions and is not reported according to the CONSORT guidance" - https://pilotandfeasibilitystudies.qmul.ac.uk/introduction/

The study still includes descriptions which present it as if it were a pilot study, e.g. the title, abstract, discussion, without reporting the requisite feasibility outcomes. I would recommend that the authors review the explanations and references provided in the link above to help with clarifying these issues.

One additional minor comment is that Table 4 could be improved by adding a column withe the mean difference and 95% confidence intervals for the differences. Additionally, the 50%EN and 100%EN columns should show the means and standard errors rather than the means and ranges - this is the standard format for reporting tables of main results for statistical inference.

7. PLOS authors have the option to publish the peer review history of their article (what does this mean?). If published, this will include your full peer review and any attached files.

Reviewer #3: No

---

## [Author Response · Author response to Decision Letter 1]

21 Jul 2020

The authors would again like to thank Reviewer #3 for insightful comments. The study is correctly identified as a small clinical trial, the word “pilot” (inappropriately) used to indicate the exploratory nature of the trial design and focus on physiological endpoints. The manuscript has been revised to avoid any ambiguity in this regard. We are aware of the limitations of a small and heterogenous sample, and hope that the reviewer feels that this has been adequately addressed in the discussions section and previous replies to the reviewers. 

Table 4 has also been revised according to recommendations.

---

## [Decision Letter · Decision Letter 2]

18 Sep 2020

Whole body protein kinetics in critically ill patients during 50 or 100% energy provision by enteral nutrition: A randomized cross-over study

PONE-D-20-06211R2

Dear Dr. Sundström Rehal,

We’re pleased to inform you that your manuscript has been judged scientifically suitable for publication and will be formally accepted for publication once it meets all outstanding technical requirements.

Kind regards,

Juan J Loor

Academic Editor

PLOS ONE

Additional Editor Comments (optional):

Reviewers' comments:

Reviewer's Responses to Questions

**Comments to the Author**

1. If the authors have adequately addressed your comments raised in a previous round of review and you feel that this manuscript is now acceptable for publication, you may indicate that here to bypass the “Comments to the Author” section, enter your conflict of interest statement in the “Confidential to Editor” section, and submit your "Accept" recommendation.

Reviewer #3: All comments have been addressed

2. Is the manuscript technically sound, and do the data support the conclusions?

Reviewer #3: (No Response)

3. Has the statistical analysis been performed appropriately and rigorously? 

Reviewer #3: (No Response)

4. Have the authors made all data underlying the findings in their manuscript fully available?

Reviewer #3: (No Response)

5. Is the manuscript presented in an intelligible fashion and written in standard English?

Reviewer #3: (No Response)

6. Review Comments to the Author

Reviewer #3: When reporting the means and standard errors in Table 4, please report them as mean (SE), without the ± as this implies a range.

7. PLOS authors have the option to publish the peer review history of their article (what does this mean?). If published, this will include your full peer review and any attached files.

Reviewer #3: No

---

## [Editor Report · Acceptance letter]

25 Sep 2020

PONE-D-20-06211R2 

Whole-body protein kinetics in critically ill patients during 50 or 100% energy provision by enteral nutrition: A randomized cross-over study. 

Dear Dr. Sundström Rehal:

I'm pleased to inform you that your manuscript has been deemed suitable for publication in PLOS ONE. Congratulations! Your manuscript is now with our production department. 

Kind regards, 

on behalf of

Dr. Juan J Loor 

Academic Editor

PLOS ONE